# Endoplasmic Reticulum Stress Induces Vasodilation in Liver Vessels That Is Not Mediated by Unfolded Protein Response

**DOI:** 10.3390/ijms25073865

**Published:** 2024-03-30

**Authors:** Sergejs Zavadskis, Anna Shiganyan, Andrea Müllebner, Johannes Oesterreicher, Wolfgang Holnthoner, Johanna Catharina Duvigneau, Andrey V. Kozlov

**Affiliations:** 1Ludwig Boltzmann Institute for Traumatology, The Research Center in Cooperation with AUVA, Austrian Cluster for Tissue Regeneration, Donaueschingenstraße 13, 1200 Vienna, Austria; sergejs.zavadskis@trauma.lbg.ac.at (S.Z.);; 2Department of Biological Sciences and Pathobiology, Institute of Medical Biochemistry, University of Veterinary Medicine, Veterinärplatz 1, 1210 Vienna, Austria

**Keywords:** endoplasmic reticulum stress, unfolded protein response, ex-vivo model, vascular tonus, Ca^2+^, NO, nitrodilators

## Abstract

There is a growing body of evidence that ER stress and the unfolded protein response (UPR) play a key role in numerous diseases. Impaired liver perfusion and ER stress often accompany each other in liver diseases. However, the exact impact of ER stress and UPR on the hepatic perfusion is not fully understood. The aim of this study was to disclose the effect of ER stress and UPR on the size of liver vessels and on the levels of Ca^2+^ and nitric oxide (NO), critical regulators of vascular tonus. This study was carried out in precisely cut liver tissue slices. Confocal microscopy was used to create 3D images of vessels. NO levels were determined either using either laser scan microscopy (LSM) in cells or by NO-analyser in medium. Ca^2+^ levels were analysed by LSM. We show that tunicamycin, an inducer of ER stress, acts similarly with vasodilator acetylcholine. Both exert a similar effect on the NO and Ca^2+^ levels; both induce significant vasodilation. Notably, this vasodilative effect persisted despite individual inhibition of UPR pathways—ATF-6, PERK, and IRE1—despite confirming the activation of UPR. Experiments with HUVEC cells showed that elevated NO levels did not result from endothelial NO synthase (eNOS) activation. Our study suggests that tunicamycin-mediated ER stress induces liver vessel vasodilation in an NO-dependent manner, which is mediated by intracellular nitrodilator-activatable NO store (NANOS) in smooth muscle cells rather than by eNOS.

## 1. Introduction

The liver produces large amounts of secretory proteins in the endoplasmic reticulum (ER) of hepatocytes. Apart from the synthesis, folding, and secretion of proteins, the ER is an important site for the intercellular calcium homeostasis [1] and subsequent calcium dependent regulatory pathways. Conditions that disrupt ER homeostasis create a cellular state commonly referred to as ‘ER stress’. There is a growing body of evidence that ER stress plays a key role in numerous physiological and pathological processes [2].

Induction of ER stress causes disturbances in protein synthesis, calcium homeostasis, and triggers the unfolded protein response (UPR), which is a cellular adaptive mechanism aiming at the recovery of ER function. The UPR has three branches, which are characterized by the elevation of stress proteins IRE1, ATF6, and PERK [3]. IRE1 catalyses the splicing of XBP1, but it also interacts with other branches. For instance, the inhibition of the PERK pathway can be involved in an activation of the IRE1 pathway and splicing of XBP [4,5], while ATF6 elevates splicing by increasing levels of XBP1 [6]. Recent advances in the understanding of the regulation of UPR signalling have shown that it interferes with numerous signalling pathways [7]. ER stress has been implicated in numerous different human diseases [2,8].

Particularly important is the role the ER stress plays in liver pathophysiology. The well-functioning ER is required in liver for synthesis of clotting factors, numerous other proteins for metabolism of drugs and toxins. Hepatocytes, the predominant cell type in the liver, are enriched in the ER and they are responsible for the majority of liver functions. Therefore, most liver disorders are accompanied by ER stress [9]. In our previous study, we observed that also haemorrhagic shock (HS) induces sustained ER stress in the rat liver [8]. The liver is one of the organs affected foremost and instantly by haemorrhagic shock, causing dramatic changes in blood flow [10]. Other pathologies that are associated with a compromised liver perfusion also induce ER stress. These pathologies include liver tumours, liver trauma, and liver cirrhosis [11]. The frequent appearance of perfusion disorders affecting the liver may be explained by the fact that it has a unique dual blood supply, which originates from the hepatic artery (25%) and the portal vein (75%). Although impaired liver perfusion and ER stress often accompany each other, the exact impact of ER stress and UPR on the hepatic perfusion is not fully understood [12].

It is known that ER stress is capable of influencing vascular homeostasis [13], which could play a pivotal role in influencing impaired organ perfusion upon shock. Several reports suggest that ER stress can be associated with hypertension and vascular remodelling [14]. It has been shown that the induction of ER stress in renal arteries reduces endothelium-derived relaxing factors and contracting factor (EDCF)-mediated contractions [15]. In the case of hypovolemia, blood flow is predominantly redirected to vital organs, such as liver, through a process known as ‘centralized’ circulation, which is believed to be achieved exclusively via peripheral vasoconstriction. The contribution of blood flow through vital organs is less well understood.

Taking into account the evidence described above, we assumed that induction of the ER stress could affect liver perfusion. It is very difficult to address this question in vivo, because of systemic influences and the inability to visualize biochemical changes in single vessels within entire organs like the liver. These aspects, however, can be explored in a model of precisely cut liver slices, which preserve both the liver’s structure and cellular functions, but it is free of systemic influences.

The general aim of this study was to understand whether activation of ER stress in the liver could influence liver perfusion. Specifically, we examined how tunicamycin, an inducer of ER stress, and inhibitors of UPR branches impact the signalling molecules regulating vascular tonus (Ca^2+^ and nitric oxide) and the single vessel volumes in precisely cut rat liver slices.

## 2. Results

### 2.1. Tunicamycin Induces Protein Aggregation Characteristics for ER Stress

Firstly, we examined whether the liver tissue in our experimental model is capable of responding to the ER stress-inducing agent, tunicamycin. The liver tissue slices were stained with Thioflavin-T, a fluorescent probe, known for its selective binding to amyloid fibrils, which are indicative of protein aggregation [3], a typical sign for ER stress (Figure 1).

The data presented in Figure 1 clearly show that tunicamycin induces ER stress. In order to test whether the induction of ER stress activates UPR in our model, we determined an early marker of UPR, the XBP1 splicing. XBP1 is a segment of IRE1 branch of UPR, but its splicing can be modulated by other branches of UPR [16]. Due to this fact, we determined effect of different UPR branch inhibitors on the XBP1s.

### 2.2. Activation and Inhibition of UPR Pathways

To accomplish this, we employed novel inhibitors specific to distinct UPR pathways: MKC8866 targeting IRE1, GSK2606414 (GSK414) targeting PERK, and Exendin-4 (Exen4) targeting ATF-6. We observed that the increased XBP1 splicing induced by tunicamycin (Tun) was abolished by MKC8866 (an inhibitor of IRE1) and Exen4 (an inhibitor of ATF-6), while *GS*K414 augmented XBP1s, as depicted in Figure 2.

Since ER stress disrupts the Ca^2+^ homeostasis, which is a key regulator of vascular tonus, we investigated intracellular levels of Ca^2+^ in response to tunicamycin and compared the effects with two well-known vasotropic agents, acetylcholine (Ach) and phenylephrine (Phn), a vasodilator and vasoconstrictor, respectively. To achieve this, we employed FLUO-4 AM calcium-sensitive staining.

### 2.3. Vascular Calcium Levels

As expected, the addition of the vasodilator Ach led to a decline in calcium levels. Likewise, the addition of the vasoconstrictor Phn resulted in an increase in calcium levels. The application of tunicamycin resulted in a reduction in calcium signal levels, similarly to that observed with acetylcholine, as illustrated in Figure 3.

Since the vasodilatory effects of acetylcholine are mediated by NO, we subsequently examined NO levels in liver slices treated with Ach, Phn, and tunicamycin. To accomplish this, we used DAF-FM, a nitric oxide-sensitive fluorescent dye. The images were taken 90 min after treatments.

### 2.4. Vascular Nitric Oxide Levels

The data presented in Figure 4 show that treatment with Ach and tunicamycin elevate the levels of NO in the vessel wall, while Phn does not change NO levels. These observations strongly suggest that tunicamycin modulates the intracellular Ca^2+^ via NO in a similar way as acetylcholine in our model. Therefore, we hypothesized that tunicamycin acts as a vasodilator. To validate this hypothesis, we assessed the effect of tunicamycin on changes of the vessel lumen (area).

### 2.5. Vessel Volume Changes in Response to Tunicamycin

Employing confocal microscopy, we captured 3D images of the vessels through 25 sections both prior to treatment, and at 45 min and 90 min after treatment, which rendered a three-dimensional image, as depicted in Figure 5a. The determination of changes in the vessel lumen was accomplished utilizing ImageJ. Only those vessel cuts which featured entire (or complete) vessel rings were taken into consideration; partial or damaged vessels were omitted from the analysis.

The data shown above suggest that tunicamycin has indeed a vasodilatory effect in liver vessels, which is mediated by an increase in intravascular NO generation. Increased NO generation upon tunicamycin treatment was additionally confirmed by elevated NO levels occurring in the incubation medium (Figure 6b). We next hypothesized that one specific branch of UPR triggers NO release, which mediates the vasodilatory effect of tunicamycin.

### 2.6. Selective UPR Pathway Inhibition

To prove this hypothesis, we selectively inhibited various branches of the UPR and assessed their influence on the vasodilative effect of tunicamycin (Figure 6). We found that tunicamycin-induced vasodilation was not affected by selective UPR inhibitors, suggesting that it is not mediated by UPR (Figure 6a). Further, NO generation was also not modulated by UPR inhibitors (Figure 6b). Our data thus suggest it is the tunicamycin-induced ER stress that targets an NO-generation mechanism. In the vasculature several NO-generation mechanisms exist. The best known is the classical mechanism of Ach action, which involves NO generation by endothelial NOS (eNOS) of endothelial cells. In addition, several endogenous NO donors are located in smooth muscle cells, also capable of regulating vascular tonus. 

Carefully analysing the images obtained in vessels treated with Ach and with tunicamycin, we observed that in the presence of tunicamycin, the NO-mediated fluorescence was present also in the deeper layers of the vessel walls, making the walls appear thicker upon treatment with tunicamycin compared to vessels treated with Ach (Figure A2). This suggests that NO may have been also generated in the smooth muscle layer of the vessel.

### 2.7. Experiments with Endothelial Cells

In order to clarify whether tunicamycin mediates vasodilation using the same mechanism as Ach, we performed experiments with cultured endothelial cells (HUVECs). Upon treating the cells with Ach or tunicamycin, we determined NO generation by fluorescent dye, DAF-FM. To dissect the contribution of eNOS we examined the effect of L-NAME an inhibitor of NOS. The abolished NO-related fluorescence of DAF-FM in the presence of L-NAME demonstrates that eNOS is indeed the source of NO release in HUVECs. Tunicamycin did not trigger NO release of the cells. In the contrary, we observed a strong decrease in the NO-levels in response to tunicamycin (Figure A1). This demonstrates that tunicamycin in contrast to Ach operates by a different NO-generation mechanism, possibly involving the part of the vessel located below the endothelium, such as the smooth muscle cells.

## 3. Discussion

In this study, we have shown that precisely cut liver slices (PCLSs) are a suitable model to study the functionality of hepatic vasculature and its response to the different treatments, such as the effects of vasodilators, vasoconstrictors, and activation of specific pathological processes such as induction of ER stress. The advantage of this model is that it allows the accessing of the local effects of treatments without systemic influences and enables to access specific morphological structures within the liver tissue.

We had shown that tunicamycin induces ER stress in the liver slice tissue using Thioflavin T (Figure 1a,b), which stains protein aggregates and amyloid fibrils. We did not observe any differences in the staining of liver tissue that would target specific structures; the staining was similar throughout the entire liver tissue, suggesting that tunicamycin induces ER stress in all types of liver cells. To determine whether or not ER stress was activating UPR in our model, we studied splicing of XBP1. XBP1 splicing is executed by the IRE1 branch, but can be additionally modulated by PERK and the ATF6 pathway. Our observation that inhibition of the PERK pathway resulted in an increased XBP1 splicing by trend (Figure 2) supports previous studies [5,16]. The PERK pathway typically activates in later stages, when the cell is not capable of recovering from ER stress. Once PERK-ATF4-CHOP is activated, it leads to apoptosis [17]. Further, addition of Exendin-4, predominantly inhibiting ATF6, has been shown to attenuate the XBP1 splicing [4], and we also found decreasing XBP1 splicing in our model. The data presented in Figure 2 suggest that tunicamycin is potentially activating all branches of UPR in our model.

Ca^2+^ and NO are two of the most important regulators of vascular tonus. In this study, we examined effect of Ach, a well-established vasodilator, Phn, a widely used vasoconstrictor, and tunicamycin in portal vessels of the liver. Ach and Phn were employed as references for changes in Ca^2+^ (Figure 3) and NO metabolism (Figure 4) upon induction of vasodilation and vasoconstriction, respectively. The results presented in the Figure 3 revealed that both Ach and tunicamycin caused a decrease in intracellular Ca^2+^ levels, while Phn caused an increase in Ca^2+^ levels (Figure 3). These data are in line with the known mechanisms of the action of Ach and Phn, which regulate vascular tonus by modulation of intracellular Ca^2+^ levels in smooth muscle cells. Phn directly interacts with smooth muscle cells, while Ach classically involves NO generated from eNOS of endothelial cells. NO in turn diffuses into smooth muscle cells and induces vasorelaxation via the NO/cGMP signalling cascade. The data shown in the Figure 3 prompted us to assume that tunicamycin-mediated induction of ER stress also modulates Ca^2+^ metabolism in a manner similar to Ach. Determination of NO levels in the vascular wall confirmed this assumption. We demonstrated that both Ach and tunicamycin increased intracellular NO levels, while Phn did not exhibit any detectable change in the NO levels (Figure 4). These observations further support our assumption that both Ach and tunicamycin operate via NO generation. An elevated NO generation was also confirmed by the increased concentrations of NO in the incubation medium upon treatment with tunicamycin (Figure 6b). The physiological effects of both Ach and Phn in our model are in line with observations using in-vivo findings [18,19]. The similarity in the actions of the ER stress inducer tunicamycin and Ach suggests that they both induce vasodilation mediated by NO and Ca^2+^.

To quantify the change in vessel lumen and assess its response to tunicamycin, we employed laser scanning confocal microscopy to capture a comprehensive 3D image of the vessel and calculated the vessel area as described in the methods section. We observed that after 45 min of incubation with tunicamycin, the vessel area had a strong trend to increase (Figure 5a); this trend became significant after 90 min of incubation (Figure 5b). The findings illustrated in Figure 5 suggest that tunicamycin induces vasodilation in a NO-dependent manner. We assumed that this mechanism was activated by one of the branches of UPR, as previously suggested [20]. To test this assumption, we selectively inhibited various branches of the UPR and assessed their impact on the vasodilatory effect of tunicamycin and generation of NO. Given that the effects of tunicamycin were most pronounced at 90 min post-treatment, we evaluated effects of all inhibitors at this specific time point.

Surprisingly, none of the inhibitors exhibited a significant effect on the tunicamycin-mediated vasodilation (Figure 6a). Similarly, none of the inhibitors exerted a significant effect on the levels of NO released in the medium. Therefore, we could rule out UPR as mediator the vasotropic effects of tunicamycin. Instead, vasodilation may be elicited by the provoked ER stress itself. One possibility is that ER stress activates eNOS in endothelial cells, resulting in an increase in NO generation and subsequent vasodilation. This could be due to the release of Ca^2+^. The activation of Ca^2+^-dependent enzymes requires much lower levels of Ca^2+^ compared to the Ca^2+^ levels required for muscle contraction. That could be the reason why we did not see any increase in Ca^2+^ levels in response to Ach and tunicamycin.

To prove this assumption, we performed experiments with a HUVEC culture (Figure A1). The cells were treated with the same concentration of tunicamycin and incubated for 90 min. To assess the contribution of eNOS, we treated cells with the NOS inhibitor L-NAME. However, instead of the expected increase in the levels of NO, we observed a strong decrease in the NO levels, which was further completely abolished by L-NAME. We therefore can exclude involvement of eNOS generating NO. Nonetheless, the decrease in NO generation in response to tunicamycin is in line with previously reported data on the inhibition of NOS upon ER stress conditions recently reported [21].

Therefore, we questioned by which mechanism tunicamycin increased the NO generation in our portal liver vessels, which we determined using two independent methods, by NO specific fluorescent dye DAF-FM in the vessel walls and by chemiluminescence as a measure for NO products present in the incubation medium. Further, we show that this NO must have another source than the endothelial NOS. Recently, an alternative pathway of NO generation, which can induce vasodilation, was described by Liu et al. [22]. This pathway operates directly in smooth muscle cells. The authors describe a pool able to release vasodilatory NO under pathological conditions. Considering that induction of ER stress represents a stressful condition, it is well possible that tunicamycin activated this mechanism in our model as well. Indeed, when comparing the LSM images in the presence of tunicamycin and Ach, the vessel walls tended to be thicker upon treatment with tunicamycin compared to Ach (Figure A2). Therefore, our data suggest an involvement of smooth muscle cells in response to tunicamycin. We conclude that tunicamycin operates as vasodilator using NO. The mechanism by which NO is released is, however, different to Ach, as it does not involve eNOS. Instead, our findings point towards an involvement of the underlying smooth muscle cells, apparently capable of increasing NO generation in response to ER stress.

## 4. Materials and Methods

### 4.1. Animals and Respective Materials

The ex-vivo experiments were performed on Sprague-Dawley rats (390–500 g; 8–12 weeks, ♂; Janvier Labs, Le Genest-Saint-Isle, France) which were housed in pairs under controlled standard animal laboratory conditions at least for 7 days for adaptation purposes before experiments with free access to standard laboratory rodent diet and water. The rats with higher weight were used in priority. The animals were anesthetized by inhalation of 8% Sevoflurane mixture (Vapor 2000, Dräger, Austria) for at least 12 min until a complete loss of paw pinch reflex and immediately decapitated afterwards (DCAP-M, World Precision Instruments, Sarasota, FL, USA).

### 4.2. Human Umbilical Vein Endothelial Cell Isolation and Culture

Human umbilical vein endothelial cells (HUVECs) were isolated as described previously [23] under approval of the local ethics committee of the state of Upper Austria (ethics committee vote #200, 12/05/200). Cells were cultured in EGM-2 (Lonza, Walkersville, MD, USA) supplemented with 5% foetal calf serum (FCS) at 37 °C and 5% CO_2_ and used in passage 6.

### 4.3. Reagents

The reagents were sourced from the following suppliers:DAF-FM Diacetate (Cat. Nr. D23842, Thermo Fischer, Waltham, MA, USA)FLUO-4 AM (Cat. Nr. F14201, Thermo Fischer, Waltham, MA, USA).Thioflavin-T (Cat. Nr. T3516-5g, f.c. 50 µM, Thermo Fischer, Waltham, MA, USA).Acetylcholine (Cat. Nr. AA6625-10MG, f.c 300 µM, Merck/Sigma-Aldrich, Darmstadt, Germany).Phenylephrine (Cat. Nr. P1240000, f.c 300 µM, Merck/Sigma-Aldrich, Darmstadt, Germany).Tunicamycin (Cat. Nr. Cay11445-10, f.c. 200 µg/mL, Cayman Chemical, Ann Arbor, MA, USA).

### 4.4. Preparation Methods

Freshly prepared 6 mm 200 µm thick rat liver slices were produced as described in a previous publication by our group [24] and afterwards stained in DPBS according to the respective protocols supplied by the fluorescent probe manufacturers. Liver slices were incubated for 90 min using tunicamycin 200 µg/mL or DMSO 1% vol., and tunicamycinin combination with UPR inhibitors, 20 µM for MKC8866, 1 µM for GSK414, and 100 nM for Exendin-4.

### 4.5. Determination of XBP1 Splicing

Total RNA from liver slices shock frozen in liquid nitrogen was isolated using RNAspin Mini Kit (Cytiva, Marlborough, MA, USA) according to the manufacturer’s instructions. The RNA was quantified spectrophotometrically at 280 nm and the purity was assessed by ratio of absorbance at 260/280 nm using a Tecan Spark plate reader (Tecan Group Ltd., Männedorf, Switzerland). cDNA was synthesized from 1 µg of total RNA using Superscript^TM^ II reverse transcriptase (200 U/reaction; Invitrogen; Carlsbad, CA, USA) and anchored oligo dT primers (3.5 μmol/L final concentration). XBP1 mRNA of unspliced (XBP1u) and spliced variant (XBP1s) were amplified by means of PCR. Each PCR reaction contained SYBR R green I^®^ (0.5×, Sigma Aldrich, Vienna, Austria), iTaq^TM^ DNA polymerase^TM^ (25 U/L; Bio Rad, Hercules, CA, USA), oligonucleotide primers (250 nmol/L each, Invitrogen; Carlsbad, CA, USA), dNTP (200 μmol/L each), and MgCl_2_ (3 mmol/L). Primer pairs for XBP1 were described elsewhere [25]. Randomly assigned no-reverse transcriptase controls corresponding to ~15% of all the samples investigated, a no-template control, and an internal standard, a pool generated of equal aliquots of cDNA of all samples investigated, were included in each measurement. Analyses were run in duplicate on a CFX96^TM^ real-time cycler (Bio-Rad, Hercules, CA, USA). The PCR products of XBP1u (196 bp) and XBP1s (170 bp) mRNA were separated on a 2% Agarose Gel, stained with peqGreen (PeqLAB, Erlangen, Germany) and visualized by UV transillumination at 300 nm. Densidometric analyses were performed using the public domain image analyses software Scion Image Version Alpha 4.0.2.3 (Scion Corp., Frederick, MD, USA). Results were used to calculate the ratio of XBP1s to XBP1u mRNA.

### 4.6. Determination of NO Levels

The NO levels in the medium were measured with Sievers nitric oxide NOA-280i analyser (Analytix, Boldon, UK). The major product of NO oxidation in the medium is nitrite (NO_2_^−^). A reducing agent (NaI) is used to convert and measure nitrite back to NO.
I^−^ + NO_2_^−^ + 2H^+^ → NO + 0.5I_2_ + H_2_O

Afterwards, the NO was quantified based on the chemiluminescent reaction with ozone by the device.

### 4.7. Microscopic Imaging and Analysis

LSM imaging was performed with an inverted confocal microscope (LSM 510, Zeiss, Oberkochen, Germany) and 10× objective. The images were acquired in ZEN 2009 (version 6.0.303; Carl Zeiss, Oberkochen, Germany) program and processed with ImageJ (version 1.53t, U. S. National Institutes of Health, Bethesda, MD, USA) software.

DAF-FM was identified as a viable option for unambiguously labelling vasculature within living tissue, as reported in the literature [26]. Our study successfully discerned a specific liver vascular component, namely the portal triad, as illustrated in Figure 7.

The analysis of 3D images involved calculation of the vessel’s volume using ImageJ software (version 1.53t) [27]. A specialized algorithm was developed, incorporating automated segmentation and computation of vessel lumen area for each image, with the objective of reducing bias originating from operator influence. In order to maintain alignment with our experimental schedule, we set a resolution of 512 × 512 pixels per scan, thereby preventing the image acquisition time from exceeding 10% of the intended experimental period.

The analysis of the vessel lumen area was performed with the following code:run(“Convert to Mask”, “method = Huang background = Dark calculate black”);run(“Despeckle”, “stack”);close(“ROI Manager”);run(“Analyze Particles...”, “size = 1500-Infinity summarize add slice”);roiManager(“Select”, 0);run(“Measure”);

Afterwards, individual areas were multiplied with respective slice thickness for volume calculation.

It is crucial to highlight that only the undamaged sections of the vessels were considered for analysis, with those exhibiting outer damage resulting from cutting procedures deliberately excluded, as illustrated in Figure 8.

## 5. Conclusions

In summary, the data obtained in this study suggest that the tunicamycin-mediated induction of ER stress results in a vasodilation of blood vessels in hepatic parenchyma. This vasodilation is NO dependent, but the source of NO is not eNOS. Our working hypothesis is that the mechanism underlying this effect is mediated by the nitrodilator-activatable intracellular NO store (NANOS), which was recently described [22]. The exact composition of this NO store is not yet clarified, but its components seem to be limited to a combination of nitrogen oxoanions such as nitrite (NO_2_^−^) and nitrate (NO_3_^−^) and S-nitrosothiols (SNOs) and that these substances are included in vesicle structures. Further studies are required to clarify mechanisms underlying the release of NO from NANOS. Although we do not know the exact mechanism of the NO release and the subsequent vasodilation, the data presented here can be important for understanding pathogenesis of diseases accompanied by circulatory failure, such as haemorrhagic shock.

## Figures and Tables

**Figure 1 ijms-25-03865-f001:**
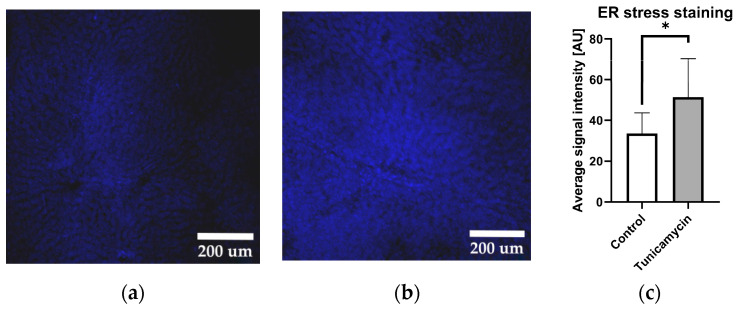
ER stress in vital liver tissue slices after 90 min visualized by staining with Thioflavin-T (10× magnification). (**a**) Typical fluorescence levels of a control tissue slice. (**b**) Representative fluorescence levels of liver tissue slices exposed to tunicamycin for 90 min. (**c**) Signal intensity quantification (whole image field) in control (treated with tunicamycin vehicle DMSO) and tunicamycin-treated samples. Unpaired *t*-test; data are presented as means ± SD; *n* = 8; * = *p* < 0.05.

**Figure 2 ijms-25-03865-f002:**
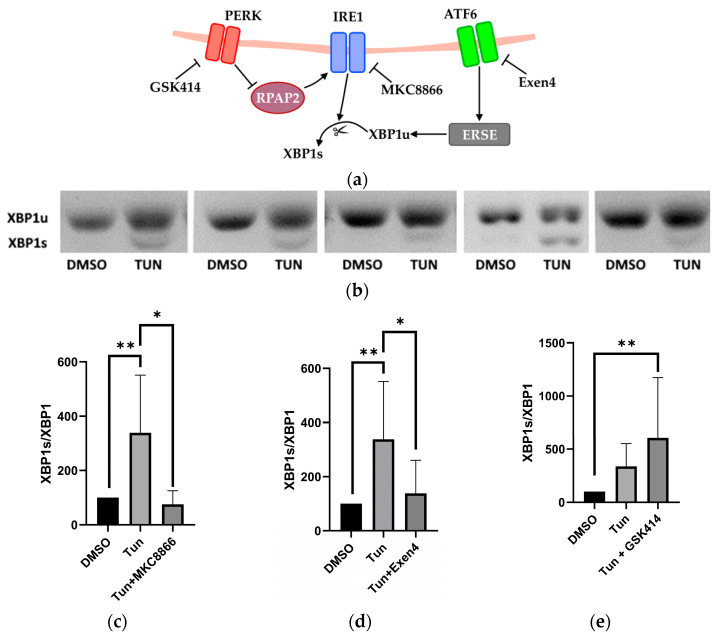
Determination of X-box binding protein 1 (XBP1) splicing as ratio between unspliced and spliced XBP1 mRNA in % of control in precision-cut tissue slices subjected to tunicamycin in combined with inhibitors targeting specific UPR pathways. (**a**) Scheme illustrating 3 UPR sensors with respective inhibitors and interactions. (**b**) Representative images of XBP1 PCR products for slices treated with DMSO (vehicle of tunicamycin) and tunicamycin-treated slices. (**c**) Simultaneous administration of the IRE1 inhibitor MKC8866 with tunicamycin results in a substantial reduction in XBP1 indicated by a reduced ratio of spliced XBP1 (XBP1s) to the unspliced XBP1 (XBP1u) amount of PCR product. (**d**) Simultaneous application of the ATF6 inhibitor Exen4 with tunicamycin reduces XBP1 splicing. (**e**) Concurrent administration of the PERK inhibitor GSK414 with tunicamycin increases XBP1 splicing. The samples were taken 90 min after treatment. One-way ANOVA; data are presented as means ± SD excluding outliers DMSO *n* = 11; Tun *n* = 11; Tun + UPR inhibitor *n* = 4/8/5; outliers were excluded with 5% ROUT; * = *p* < 0.05; ** = *p* < 0.005.

**Figure 3 ijms-25-03865-f003:**
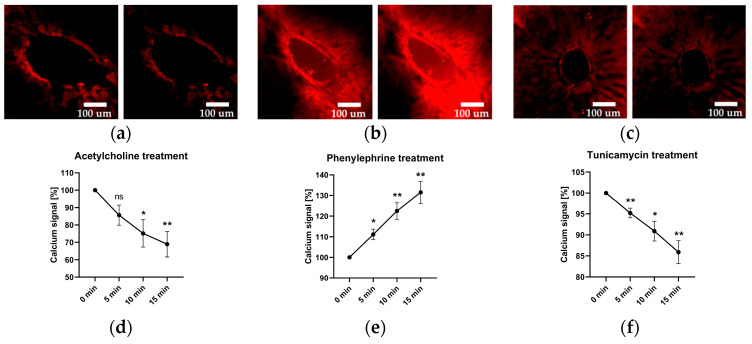
Effect of acetylcholine, phenylephrine, and tunicamycin on the intracellular calcium levels, visualized with FLUO-4 AM, f.c. 5 µM. The images are arranged in pairs, wherein the left image represents calcium levels immediately after treatment (0-min time point), while the corresponding right image represents the 15-min time point. The panels (**a**–**c**) are representative images showing the intracellular calcium levels upon treatment with acetylcholine, phenylephrine, and tunicamycin, respectively. Quantification of these changes are presented in panels (**d**–**f**) for acetylcholine, phenylephrine, and tunicamycin treatments, respectively. One-way ANOVA; data are presented as means ± SD; *n* = 4 biological replicates; *n* = 12 technical replicates; ns—not significant; * = *p* < 0.05; ** = *p* < 0.005.

**Figure 4 ijms-25-03865-f004:**
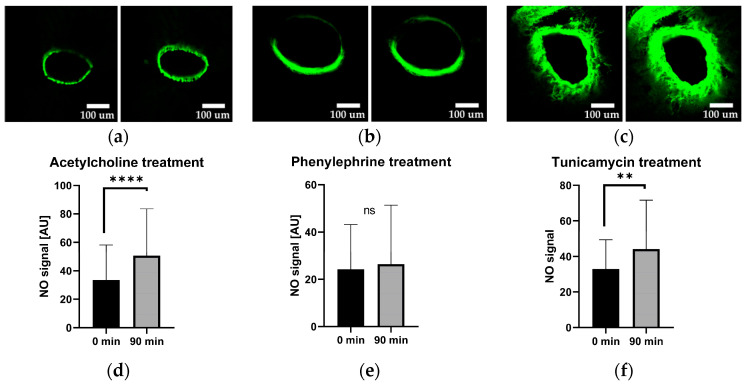
Effect of acetylcholine, phenylephrine, and tunicamycin on the intracellular nitric oxide levels, visualized with 10 µM DAF-FM. The representative images are arranged in pairs, wherein the left image depicts nitric oxide levels immediately after treatment (0-min time point), while the corresponding right image represents the 90-min time point, the scale bar represents 100 µm distance. (**a**) Representative images of NO levels following acetylcholine treatment. (**b**) Representative images of NO levels following phenylephrine treatment. (**c**) Representative images of NO levels following tunicamycin treatment. Quantification of signal changes are presented in panels (**d**–**f**) for acetylcholine, phenylephrine, and tunicamycin treatments, respectively. Statistical evaluation was carried out by Wilcoxon matched-pairs signed rank test (Shapiro–Wilk test not passed); data are presented as means ± SD; *n* = 5 biological replicates; *n* = 70 technical replicates; ns—not significant; ** = *p* < 0.005, **** = *p* < 0.0001.

**Figure 5 ijms-25-03865-f005:**
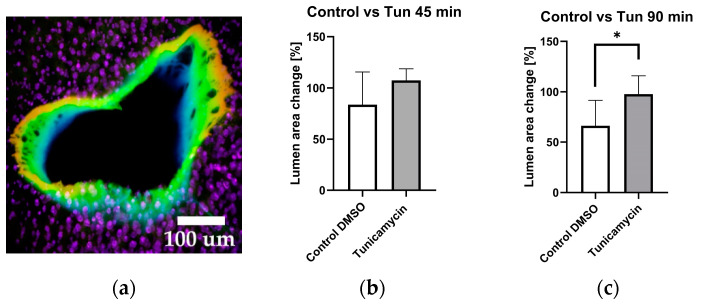
Estimation of the volumetric changes of vessels in response to tunicamycin. (**a**) Representative 3D image of an intact vessel with nuclear counterstain of surrounding hepatocytes. The liver slices were stained with DAF-FM and NucSpot650 Live for detection of NO and nuclei, respectively. (**b**) Effect of tunicamycin on blood vessel area after 45-min exposure to tunicamycin. (**c**) Effect of tunicamycin on blood vessel area after 90-min exposure to tunicamycin. Unpaired *t*-test; data are presented as means ± SD; *n* = 6; * = *p* < 0.05.

**Figure 6 ijms-25-03865-f006:**
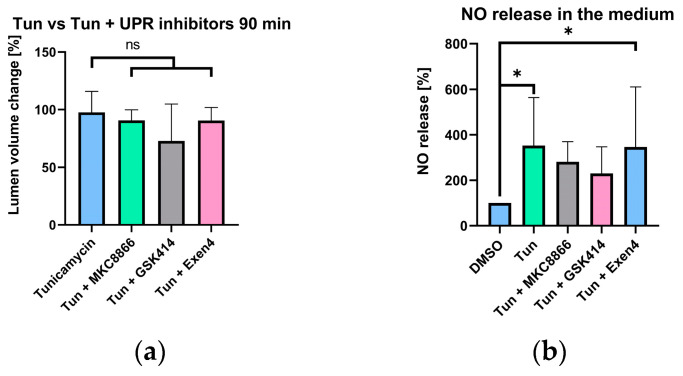
Effect of UPR inhibitors on the elevated lumen volume and NO generation induced by tunicamycin. (**a**,**b**) Effect of MKC8866, GSK414, and Exen4, the selective inhibitors of IRE1, PERK, and ATF6 pathways, respectively, on the vessel lumen volume (**a**) and on the levels of NO (**b**). *n* = 6. One-way ANOVA; data are presented as means ± SD; ns—not significant; * = *p* < 0.05.

**Figure 7 ijms-25-03865-f007:**
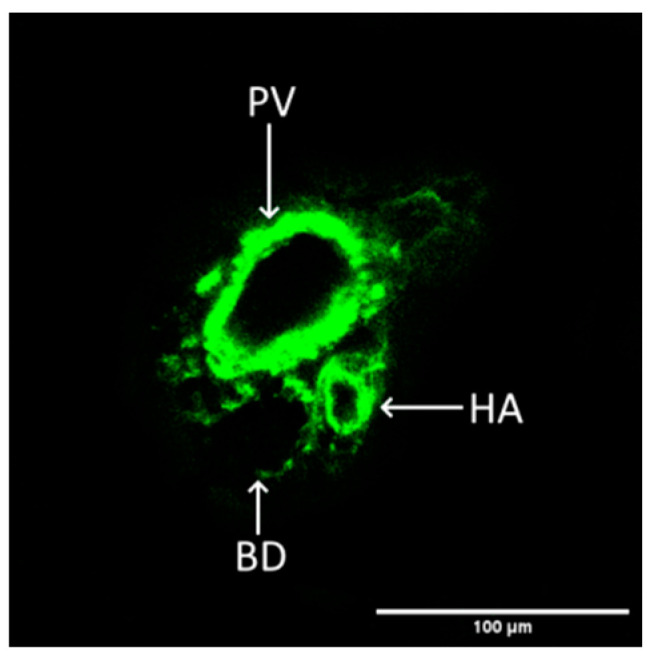
Representative image of liver vasculature depicting a portal triad, NO—sensitive staining. PV—portal vein, HA—hepatic artery, BD—bile duct.

**Figure 8 ijms-25-03865-f008:**
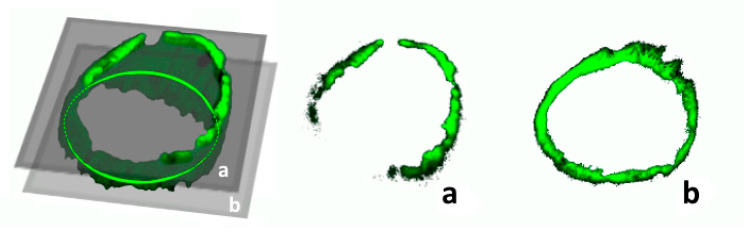
A complete 3D render of a portal vein. An example criterion for slice exclusion is images (**a**,**b**). (**b**) The top slice (1st out of 25) of the Z-stack render, excluded as lumen area calculation is not possible due to incomplete structure. (**a**) Representative image of an intact vessel suitable for lumen area calculation (15th out of 25).

## Data Availability

The data presented in this study are available on request.

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
