# Peer review of "Endoplasmic Reticulum Stress Induces Vasodilation in Liver Vessels That Is Not Mediated by Unfolded Protein Response"

_ijms, 2024, doi:10.3390/ijms25073865_

Round 1

Reviewer 1 Report

Comments and Suggestions for Authors

In this study the authors investigate the induction of endoplasmic reticulum (ER) stress and the unfolded protein response (UPR) in an ex-vivo rat liver model, focusing on vascular tonus.

While the study design seems appropriate, there are areas that need to be improved:

1)     The abstract could be more precise in describing the specific aims and outcomes of the study. For example, it could clearly state the hypothesis being tested and the key findings, rather than providing a general overview. It could benefit from briefly explaining why studying these processes in the context of vascular tonus is important or how it contributes to the broader understanding of ER stress-related mechanisms. Overall, while the abstract provides an interesting glimpse into the study's findings, it could benefit from some refinements to enhance its clarity and impact.

2)     The introduction provides a comprehensive overview of the role of endoplasmic reticulum (ER) stress and the unfolded protein response (UPR) in liver physiology and pathology, particularly in the context of hepatic perfusion disorders. Here are some comments to improve its clarity and structure:

-  Some sections could be more clearly articulated. For example, the discussion on the UPR branches and their interactions could be simplified for better readability.

- Other sections contain detailed information that could be condensed to improve readability. For instance, the discussion on the liver's dual blood supply and its relevance to perfusion disorders could be summarized more succinctly.

- Could be better connect the background information to the study's rationale and objectives. This would help readers understand how the study fits into the broader context of ER stress and liver physiology. As example look at: https://doi.org/10.1016/j.ymgme.2023.107700

3)     In the section results, as they are well analyzed, I have some doubts about the Figures presented, in particular:

-  Figure 1 has low definition and poor quality'. It is difficult to interpret.

- Figure 3 e Figure 4, as shown above for Figure 1, are of low quality and not interpretable. Both appear to be the same acquisition, with different exposure during the acquisition. Moreover, in Figures 3 and Figures 4 there is an absence of appropriate and uniform scale bars.

- In the Figure 5 lacks scale bars.

I suggest a new acquisition with the integration of multiple and different acquired fields to add in supplementary material (at least 5), with a standardization of scale bars.

4)      Although the discussion is critical and well written, I am not convinced by the form. Paragraph splitting makes reading less smooth for the reader. I suggest that you standardize in one paragraph and depart the conclusions in a separate paragraph.

Based on the above comment, in my opinion this manuscript could be accepted for the publication in our journal after a major revision.

Author Response

Please, see attachment

Reviewer 2 Report

Comments and Suggestions for Authors

The authors present an article on induction of ER stress with tunicamycin and activation of UPR within an ex-vivo rat liver vital tissue slice model, focusing on vascular tonus.

It is an ex-vivo experimental study with possible in-vivo applications in patients with liver dysfunction. Basic research is essential for further translational research to bring the advances to daily clinical practice. I must congratulate the authors for the quality of this study. The tables and graphs presented are very good.

It is strange to find the material and methods section after the results and discussion section, I recommend placing it after the introduction to help the better understanding of the study.

The manuscript authored by Zavadskis et al. delves into the relationship between ER stress and the vasodilation of portal vasculature. The experiments conducted are meticulously designed and executed, and the conclusions drawn are well substantiated by the results obtained. Based on this thorough evaluation, the article is deemed worthy of publication.

The objective of this study is to elucidate whether the activation of ER stress in the liver induces vasodilation in the hepatic vasculature and, if so, to delineate the underlying mechanisms. This investigation employs an ex-vivo tissue slice model to examine the hepatic vasodilatory effects of ER stress induced by tunicamycin.

Experiments have been performed that examine the role of ER in hypovolaemia or renal impairment, but the exact role of ER stress in various liver diseases and their contribution to hepatic perfusion disorder is not fully understood. This article represents a novelty in this field.

The methodology of the study is adequate, as it is based on an ex-vivo experimental model. The steps leading to the results are satisfactorily explained, so it can be considered a reproducible experiment.

The conclusions support and summarise the results described in the manuscript. The limitations of the study are also adequately discussed. This experiment may open the door to future in vivo studies that benefit patients, which is after all the goal of medical research.

References are adequately cited, refer to studies of the last few years, and do not present errors in nomenclature, following the standards of the journal.

This is an article of high scientific quality, which is also conveyed by the data presented in the Figures and Tables. I have no suggestions beyond those considered by the Editors of the journal.

Author Response

Please, see attachment.

Reviewer 3 Report

Comments and Suggestions for Authors

The manuscript titled "Endoplasmic Reticulum Stress Induces Vasculature Expansion..." is a very good section of the paper. Essentially, apart from formal control, which is not available in its application.

Below are my suggestions for changes and comments:

  1. The title seems a bit long and exhaustive for understanding. Perhaps it could be slightly refined.
  2. Is the abstract too long? The last three sentences in the abstract could be rewritten as they don't sound good. Please clarify the aim of the study.
  3. Why are "Tunicamycin" and "NO Signaling" included as keywords? Could the last two be somehow combined?
  4. The last two paragraphs need to be merged for clarity and to express the aim of the study more explicitly.
  5. Figure 1, panel c - could be enlarged as it is currently not visible but important.
  6. Panel B in Figure 2 could be enlarged. Also, why is there a gap before panels c, d, and e?
  7. The same goes for Figures 3 and 4, panels d, e, f. They are too small and not easily visible.
  8. "Control DMSO" - do the authors have any influence on the effects of DMSO in this type of study?
  9. "In summary of the obtained data..." - perhaps it would be worth separating this section as Conclusions?

It's important to note that this work is very good, enjoyable to read, and significant. Please consider these suggestions for a minor revision.

Author Response

Please, see attachment

Reviewer 4 Report

Comments and Suggestions for Authors

The presented manuscript by Zavadskis et al. focuses on the ER stress and its influence on vasodilation of portal vasculature. The set of experiments is well planned and performed, the conclusions are supported by obtained results. The article deserves to be published.

The Authors focused on the induction of endoplasmic reticulum (ER) stress in HUVEC cells and an ex-vivo rat liver model, to define the intracellular mechanisms involved.

The Authors explained the role of ER stress and UPR in vasodilation of liver vessels and proved that vasodilatory effect is not mediated neither by UPR nor by eNOS activity.

Indirect confirmation that mechanisms other than eNOS activation pathway must be involved in vasodilation, the role of releasing NO from smooth muscle stores is assumed.

In my opinion, the metodology is fine.

The Authors conclude from the results obtained that induction of ER stress leads to vasodilation, but the process is not dependent on eNOS activity, what was proved in the in vitro model of HUVEC cells.

In my opinion all the presented Figures are fine.

Author Response

Please, see attachment.

Round 2

Reviewer 1 Report

Comments and Suggestions for Authors

Dear authors,

Although your manuscript has improved considerably, in line with the requests of the reviewers, the images presented still leave me with some doubts.

I had requested the acquisition of new images because they did not seem to me qualitatively good.

The authors have added a Scale Bar to previously acquired images, which suggests that they have been modified following our requests. I don’t think that’s entirely correct.

Anyway, I’ll leave the final decision to the publisher.
Best wish.

Author Response

Please, see attached file.
